# Impact of the Cancer Cell Secretome in Driving Breast Cancer Progression

**DOI:** 10.3390/cancers15092653

**Published:** 2023-05-08

**Authors:** Syazalina Zahari, Saiful Effendi Syafruddin, M. Aiman Mohtar

**Affiliations:** UKM Medical Molecular Biology Institute (UMBI), Universiti Kebangsaan Malaysia, Kuala Lumpur 56000, Malaysia; p111431@siswa.ukm.edu.my (S.Z.); effendisy@ppukm.ukm.edu.my (S.E.S.)

**Keywords:** metastasis, secretome, immune modulation, tumor microenvironment, drug resistance, therapeutic targets, precision oncology

## Abstract

**Simple Summary:**

Breast cancer is a complex disease that remains a significant public health challenge. The breast cancer cells secrete various substances collectively known as the secretome, which include proteins, lipids, and nucleic acids that contribute to the growth and spread of breast cancer. The secretome plays a crucial role in the development and progression of breast cancer by modifying signaling pathways and creating an environment supporting cancer growth while evading the immune system. Additionally, the secretome is responsible for the development of resistance to cancer drugs, making it a significant challenge for effective treatment. Therefore, understanding the role of the secretome in breast cancer is essential for developing innovative therapies. This review provides insights into the impact of the secretome on breast cancer progression and its interactions with the tumor microenvironment, and explores potential therapeutic opportunities targeting the secretome components. By identifying specific molecules and signaling pathways involved in the secretome, new targets for therapeutic intervention can be identified, which can ultimately improve outcomes for breast cancer patients.

**Abstract:**

Breast cancer is a complex and heterogeneous disease resulting from the accumulation of genetic and epigenetic alterations in breast epithelial cells. Despite remarkable progress in diagnosis and treatment, breast cancer continues to be the most prevalent cancer affecting women worldwide. Recent research has uncovered a compelling link between breast cancer onset and the extracellular environment enveloping tumor cells. The complex network of proteins secreted by cancer cells and other cellular components within the tumor microenvironment has emerged as a critical player in driving the disease’s metastatic properties. Specifically, the proteins released by the tumor cells termed the secretome, can significantly influence the progression and metastasis of breast cancer. The breast cancer cell secretome promotes tumorigenesis through its ability to modulate growth-associated signaling pathways, reshaping the tumor microenvironment, supporting pre-metastatic niche formation, and facilitating immunosurveillance evasion. Additionally, the secretome has been shown to play a crucial role in drug resistance development, making it an attractive target for cancer therapy. Understanding the intricate role of the cancer cell secretome in breast cancer progression will provide new insights into the underlying mechanisms of this disease and aid in the development of more innovative therapeutic interventions. Hence, this review provides a nuanced analysis of the impact of the cancer cell secretome on breast cancer progression, elucidates the complex reciprocal interaction with the components of the tumor microenvironment and highlights emerging therapeutic opportunities for targeting the constituents of the secretome.

## 1. Introduction

Breast cancer is a disease that exhibits genetic and clinical heterogeneities with multiple cellular origins, encompassing various subtypes [1]. Breast cancer is the most diagnosed and life-threatening malignancy in women and the leading cause of cancer death in women worldwide [2]. According to GLOBOCAN 2020, the estimated prevalence of breast cancer in both sexes and for all ages is 7.79 million in 5 years, ranking number one in incidence at 2.26 million worldwide and fourth in the mortality rate [2]. In the United States, breast cancer is the second most significant cause of death by cancer among women overall, but ranks highest among Black and Hispanic women [3,4]. A localized breast cancer incidence has a good prognosis, with a five-year survival rate of more than 80% [5]. Usually, metastatic breast cancer is rare at initial diagnosis (around 6–7%). However, approximately 30% of patients diagnosed with early stages will eventually acquire recurrent or metastatic breast cancer [5,6,7]. Cases of patients with recurrent breast cancer are often fatal; survival is typically within five years of diagnosis [5].

Breast cancer has undergone several classifications over time, but the most used and widely accepted classification system of breast cancer involves the assessment of the expression of estrogen (ER), progesterone (PR), and human epidermal growth factor 2 (HER2) hormone receptors via immunohistochemical analysis. This renders breast cancer into four main subtypes: luminal A, luminal B, HER2-enriched, and basal-like [8,9]. Luminal-like breast cancer subtypes, characterized by ER and/or PR on the surface of breast cancer cells, are the most common breast cancer [10,11]. Luminal breast cancers are generally considered less aggressive than other breast cancer subtypes, such as the basal-like that do not express hormone receptors. Between the two Luminal subtypes, Luminal A breast cancers are less aggressive and have a better prognosis [12]. HER2-positive subtype overexpresses HER2, which accounts for about 20–25% of all breast cancer. Basal-like or triple-negative breast cancer (TNBC) lacks the above three key receptors. HER2 and TNBC tend to be more aggressive than other breast cancer subtypes, and are associated with a higher risk of recurrence and poorer prognosis if left untreated. Standard treatments for all subtypes would be surgical resection, radiotherapy, and chemotherapy, whereas targeted and immunotherapy would be the options to treat the specific subtypes [13,14,15]. This classification is, therefore, crucial to tailor specified treatment for the breast cancer patient. Recent therapeutic approaches have emerged, such as targeting metabolic pathways, immunotherapy, conjugated antibodies, and vaccines [16]. Therefore, it is crucial to comprehend the onset and course of breast cancer pathogenesis to create interventions that can improve cancer patients’ health and well-being.

While there have been significant advances and breakthroughs in breast cancer research, there is still much to learn about this disease. This is partly due to the complexity and heterogeneity of the disease. In recent years, breast cancer’s onset and metastatic properties have been linked to the extracellular moieties surrounding the tumor cells [17,18,19]. This includes the proteins secreted by cancer cells and other cellular constituents within the tumor microenvironment (TME). These secreted molecules released by tumor cells (termed the secretome) could influence the therapeutic response and clinical outcome, such as gaining resistance to cancer drugs and therapies, making its pathological evaluation indispensable in cancer management. Therefore, this review aims to evaluate the functional impacts of cancer cell secretome in the pathogenesis of breast cancer. While there have been several reviews linking the secretome and breast cancer progression previously, this review emphasized the recent findings, particularly on the protein molecules secreted in the tumor microenvironment and the secretome’s interaction with major components of the TME that contribute to the hallmarks of breast cancer. Last, the relevance of targeting the components of breast cancer secretome is discussed. 

## 2. The Topography of Breast Cancer Secretome

The secretome can be defined as both soluble and insoluble factors that are released or secreted into the extracellular environment. These include chemokines, cytokines, growth factors, coagulation factors, hormones, enzymes, glycoproteins, and nucleic acids. These factors can be secreted as naked components or cargo in vesicular compartments, such as extracellular vesicles (EVs). The latest human secretome atlas project highlighted that 2641 genes encode proteins predicted to be secreted in humans [20]. This number observably varies on cellular perturbation and disease development. Studies have shown that cancer cells, for example, have abnormal secretomes compared to their non-cancerous counterparts and therefore have functional impacts on cancer development [21,22,23]. The secretome is typically identified by high-throughput omics platforms, particularly protein identification by mass-spectrometry-based analysis.

The most basic and extensively researched secretome type is the cancer cell-derived conditioned medium (CM) of cancer cells grown in 2D or 3D culture [24]. Typically, serum proteins and scaffold-free (formation of spheroids without hydrogels, laminin, collagen, or ECM gel) are removed from the medium, and then culturing the cancer cells in serum-starved media in a short period (24 or 48 h). The medium is collected and centrifuged to remove apoptotic bodies, concentrated, and further subjected to secretome identification. This method’s benefits include obtaining relatively large sample sizes and comparing data quantitatively following cancer cell modification [25,26]. In an in vivo setting, on the other hand, the cancer cell secretome, including breast cancer, can be isolated from the bodily fluids of the cancer patient. Often time, for most cancer types, serum or plasma is the primary source of secretomic studies. 

In breast cancer patients, there are additional essential avenues to breast cancer research in that several localities of the breast ductal/lobular system are enriched with the secretome population. For example, the proteins can be secreted or shed by the tumor or stromal cells into the tumor interstitial fluid (TIF). This fluid, which surrounds the stromal and tumor cells, is thought to contain signaling constituents crucial for intercellular communication and the growth of the tumor. To obtain this TIF, small fragments of fresh tumor tissue are cultured in a buffered solution [27]. After centrifugation, the secretome will be released into the supernatant [28]. In addition, the secretome fractions from nipple aspirate fluid (NAF), pleural effusion (PE), stool, and ascites are other types of fluids that can be analyzed and have been previously shown to contain cancer-specific proteins as compared to the baseline patient. NAF extraction has been accomplished with varying success rates by using either a breast pump, massage, warming of the breast or combinations of these methods [29,30]. The release of NAF into the ducts could be enhanced by administering nasal oxytocin, increasing the yield of NAF in breast cancer patients [31]. It is known that breast cancer spreads into the pleural space via lymph vessels [32]. Hence, the PE sample is withdrawn from this pleural space localized between the inside of the chest wall and the outside of the lung via thoracentesis. Studies have shown that the gut microbiota induces multiple pathways linked to breast tumor growth [33,34] through endogenous estrogen regulation and systemic inflammation activation [33,35,36,37]. Therefore, stool samples are subjected to secretome studies that are usually extracted using a fecal swab test kit. Malignant ascites, a severe occurrence in cancer patients, are typically signs of late-stage cancer, particularly in those with stage IV breast cancer. Paracentesis is used to drain ascites from the abdominal cavity [38,39]. Overall, the breast components have the potential to add another essential avenue to the efforts to advance breast cancer research. 

## 3. The Crosstalk between Cancer Cell Secretome and the Tumor Microenvironment

The tissue secretome is markedly changed during cancer development compared to normal tissue. The aberrant gene mutations in cancer cells cause high protein synthesis and secretion demand. The increased secretion levels resulted in the alteration of critical processes that augment tumor growth. The release of the secretome also could modulate the cancer extracellular space, particularly the TME behavior. The TME is an ecosystem that includes a heterogenous group of invading and resident host cells within a body surrounding the tumor [40]. TME composition is complex and varies according to tumor type.

Nevertheless, its hallmark features consist of cancer stem cells (CSC), immune cells, extracellular matrix (ECM), blood vessels, and cancer-associated fibroblast (CAFs) [41,42]. In the early onset of cancer, reciprocal heterotypic paracrine signaling between tumor cells and other TME components triggers a cascade of biochemical and biomechanical changes, leading to a dynamic interaction between TME components. The prerequisite of malignancy for many solid cancers is the alteration of ECM. This involves the secretion of ECM remodeling enzymes by newly transformed tumor cells to degrade the basement membrane, which provides a conducive environment for tumor invasion. 

During malignancy, the stroma will undergo alterations to incite growth, invasion, and metastasis of cancer cells. These changes include CAF formation, which comprises a significant portion of the reactive tissue stroma and is critical in regulating tumor progression. The rearrangement of TME components via dynamic and mutual crosstalk is thought to drive tumor fitness and metastatic potentials [40,43,44]. The relationship between the components of TME also imposes a varying degree of response to therapy and drug resistance [45,46]. Most cancer types demonstrated fibrotic or rigid TME architecture [47,48]. Other TMEs have a more vascular microenvironment compacted with blood vessels [49,50]. The different architecture and variety of components of TMEs may also obscure the delivery of drugs to reach cancer cells [51]. Here, the interplay of breast cancer secretome with different members of TME is discussed.

### 3.1. Breast Cancer Cell Secretome and Stromal Components

Stromal cells are connective tissue cells found throughout the body in various organs and tissues that provide structural support and regulate cell growth, differentiation, and migration [52]. They are a diverse group of cells with different functions depending on their location, which includes fibroblasts, adipocytes, and pericytes. Stromal cells are essential in both normal tissue function and disease processes, with abnormalities in their function implicated in various diseases such as cancer, fibrosis, and autoimmune disorders [53,54]. The breast cancer stroma is a heterogeneous mixture of non-malignant cells comprising endothelial cells, lymphatic vessels, infiltrating immune cells, adipocytes, fibroblasts, and mesenchymal stem cells (MSCs) [55]. Breast cancer interacts bi-directionally with stromal cells by secreting cytokines, growth factors, and other signaling molecules. The interaction can either induce or inhibit stromal cells. Affected stromal cells then trigger the secretion of paracrine factors that promote the growth and progression of breast tumors, promoting epithelial-mesenchymal transition (EMT) and inducing invasive capabilities [56,57]. The most abundant breast cancer stroma cell is CAFs. CAFs facilitate several molecular interactions between the stroma and the breast cancer cells. For example, cancer cells secrete TGF-β and exert a paracrine effect to induce the differentiation of fibroblasts into CAFs [58]. As a result, CAFs secrete factors to promote angiogenesis and growth [59,60]. CAFs have a more proliferative capability and are aggressive compared to normal fibroblasts. CAFs are not always formed from the transformation of normal fibroblasts in the TME. Still, they may come from tissues or progenitor cells, such as stellate cells, bone marrow-derived fibrocytes, MSCs, adipocytes, pericytes, endothelial cells, and smooth muscle cells [61]. CAFs can induce the degradation of nearby ECM by secreting matrix metalloproteinases (MMPs) and urokinase-type plasminogen activators. At the same time, CAFs secrete large amounts of type I, III, IV, and V collagen, fibrinolytic protein, hyaluronic acid, and laminin to induce ECM remodeling. CAFs also secrete high amounts of growth factors such as transforming growth factor beta (TGF-β), hepatocyte growth factor (HGF), vascular endothelial growth factor (VEGF), and fibroblast growth factor (FGF) to induce EMT activation, angiogenic shift, metastasis, and metabolic reprogramming [62,63,64,65]. CAFs promote aggressive phenotypes in breast cancer by inducing the EMT by TGF-β1 through paracrine signaling [66]. The secretion of IL-1β, IL-6, IL-8, SDF-1, and NFκB by CAFs contributes to immune cell recruitment that may contribute to tumor progression, cancer survival and drug resistance by creating a protective niche in the TME [67,68].

Another cellular element that makes up breast cancer TME is adipocytes. Recent findings suggest that they promote the advancement of the tumor through a reciprocal and constantly evolving interaction with the cancerous cells and TME [69,70]. For example, breast cancer cells secrete pro-inflammatory cytokines such as IL-6 and tumor necrosis factor (TNF-α) that can promote adipose tissue inflammation and disrupt normal adipose tissue function [71]. In addition, adipokines such as leptin, adiponectin, autotaxin, and resistin are hormones produced by adipocytes that regulate metabolism and energy homeostasis [72]. In the TME, breast cancer cells can secrete adipokines such as adiponectin and leptin, promoting tumor growth and invasion [73].

### 3.2. Breast Cancer Cell Secretome and Immune Modulation

Breast cancer cells can secrete various factors such as cytokines, chemokines, growth factors, and EVs, which can modulate the immune response and promote tumor growth. Studies have shown dynamic interaction between tumor cells and tumor-infiltrating inflammatory cells such as lymphocytes (TILs), plasma cells, dendritic cells, macrophages, and neutrophils [74,75]. One of the mechanisms by which breast cancer cells modulate the immune response is by secreting cytokines such as interleukin 6 (IL-6), which can activate immune cells such as T cells, B cells, and macrophages [76,77]. IL-6 can also induce the production of other cytokines, such as IL-10 and transforming growth factor-beta (TGF-β), which can suppress the immune response and promote tumor growth [78,79]. Furthermore, breast cancer can recruit tumor-infiltrating lymphocytes (TILs) that are particularly abundant in ER and PR-negative or HER2-enriched cancers [80,81,82,83]. TILs in breast cancer are primarily T cells, with very few B cells [84,85]. Different types of T cells have varying effects on the TME. For example, CD8+ cytotoxic T cells kill tumor cells by secreting granzyme and perforin, which is mediated by interferon γ (IFN-γ). Type 1 helper T cells are induced by IFN-γ and IL-12 signals and activate antigen-presenting cells (APC) for effective CD8+ differentiation and progression [86,87]. Type 2 and type 17 helper T cells play more diverse roles in advancing breast cancer [88,89]. Follicular helper T cells perform critical functions in antigen-specific B cell maturation, increasing local memory cell differentiation and supporting the establishment of tertiary lymphoid organs, enhancing local anti-tumor immune response [90,91]. Regulatory T cells are essential for homeostasis and tolerance of the immune system. Their involvement in TME enhances immunosuppression via immunosuppressive cytokines (IL-10, TGF-β) and direct cell–cell contact suppression [92]. In general, the existence of type 1 helper T cell response is associated with a better prognosis, while regulatory T cells can aid in the progression of breast cancer [93,94]. B cells and T cells can be seen in close proximity within the TME, notably at tertiary lymphoid structures, and their presence is thought to have predictive value [85,95]. 

Dendritic cells are the most prominent APC that deliver antigens to T cells, including tumor-derived antigens [96]. High dendritic cell infiltration in breast cancer is associated with poor prognosis as it was shown to prevail metastasis via CXCR4/SDF-1 chemokine axis [97]. However, a high subset of dendritic cells in TNBC is also associated with better disease-free and disease-specific survival [98]. Tumor-associated macrophages (TAMs) are tumors’ most common innate immune cells. An in vivo study on breast cancer bone metastasis revealed that bone metastases were markedly inhibited by macrophage ablation [99]. Furthermore, it was discovered that IL4R-dependent monocyte-derived macrophages drive bone metastases in breast cancer. Neutrophils are increasingly becoming identified as immune cells that infiltrate tumors. IFN-γ and IFN-β exposure generates N1 pro-inflammatory and anti-tumor of TAN, while TGF-β exposure induces N2 anti-inflammatory response and pro-tumor TAN [100,101]. In breast cancer mouse models, TAN can inhibit CD8+ proliferation and promote the recruitment of immunosuppressive cells in TME, but its effects in human patients are yet to be explored [102]. 

Breast cancer cells can also secrete EVs such as exosomes, which are small membrane-bound vesicles involved in cell-to-cell communication. Exosomes encapsulate various factors that can modulate the immune response and promote tumor growth. In breast cancer, exosomes carry miRNAs as one of their cargos that can target genes involved in immune regulation, suppressing the immune response and promoting tumor growth [103]. Exosomes can also carry proteins such as programmed death-ligand 1 (PD-L1), a checkpoint inhibitor involved in immune evasion by tumor cells. PD-L1 can bind to PD-1 receptors on T cells and suppress the immune response by inhibiting T cell activation and promoting T cell apoptosis [83,104]. This allows the cancer cells to evade the immune system and proliferate uncontrollably. 

### 3.3. Breast Cancer Cell Secretome and Metastasis

Tumor metastasis is a multi-step process comprising local invasion, intravasation, migration through the lymphatics or arteries, extravasation, and colonization, giving rise to metastases in distant organs [105]. Organ-specific colonization mainly depends on the dynamic and mutual interrelationship between tumor cells and the distant/secondary cells/organs secretome, as well as the components of TME [106]. Studies indicate that the genetic alterations found in breast cancer cells that have spread to the bone marrow are often different from those in primary tumors, and metastatic locations may be influenced by multiple microenvironments and cellular and molecular processes [107]. Different cancer types typically spread to multiple but preferred organ sites. Breast cancer, for example, has its preferential metastatic sites, including bones, lungs, liver and brain (Figure 1) [108]. All breast cancer subtypes can result in bone metastases, with the luminal A subtype posing a higher risk for bone recurrence, and luminal B patients are more likely to experience bone metastasis as the first site of relapse [109].

In contrast, the incidence of bone metastasis is higher in luminal subtype tumors compared to other subtypes. Additionally, lung-specific metastasis is more common in luminal B and basal-like subtypes, while liver metastasis is more frequently observed in patients with HER2-enriched subtypes [109,110]. The basal-like subtype is more prone to spread to distant lymph nodes, the brain, and the lungs than to the liver and bones [111]. Through a complex system involving interaction between the stromal components of the primary tumors and organs, a pre-metastatic niche or a suitable microenvironment can be created in secondary tissues or organs before the occurrence of metastases [112]. It was reported that mobilized tumor bone marrow-derived cells (BMDCs) influence creating a favorable milieu for metastatic lung colonization [113]. The proteins secreted by the primary tumor, VEGF, and placental growth factor (PIGF) affect the bone marrow mesenchymal stem cells, causing the BMDCs to go to the preferred site for metastasis before the disseminated tumor cells do. During the process of EMT, ZEB1/2, SNAIL, GOOSECOID, and FOXC2 are upregulated, resulting in the loss of epithelial cell polarity and gap junction activities [62]. Breast cancer cells can alter the composition and organization of the ECM by secreting enzymes such as matrix metalloproteinases, cathepsins, and plasminogen activators, which break down ECM proteins and create space for tumor growth [114,115,116].

In contrast, during the mesenchymal-to-epithelial transition (MET), tumor cells regain their epithelial properties and interact through juxtacrine signaling (Notch and Wnt factors), colonizing the metastatic site. EMT significantly alters the dynamic environment of TME, drawing stromal and immune cells from different tissues to the TME through cytokines (such as TGF-β) and chemokines (such as CXCL2, CCL22, MMP, IL-6, and IL-8) released by breast cancer cells. It is worth noting that different breast cancer subtypes are involved in various pathways that mediate EMT and are regulated by diverse cell signaling mechanisms [117].

Endothelial cells create a continuous barrier in most organs, including the brain, which prevents cancer cells from freely entering. By establishing interactions with tumor cells via L-or-P-selectin secreted by endothelial cells, platelets and white blood cells can assist tumor cells in moving through the vasculature [118,119]. As a result, higher expression of selectin ligands by tumor cells is strongly associated with metastatic progression and poor prognosis. Moreover, TGF-β or small mother against decapentaplegic (SMAD) signaling pathway, has been found to increase cancer cells’ retention in the lungs and give breast cancer cells the ability to damage the capillary wall of the lung and develop lung metastases [120]. In addition, target tissues where cancer cells can metastasize or migrate can secrete chemokines that cause directed cell migration, trigger signaling cascades and keep track of cytoskeletal rearrangement and adhesion [121].

### 3.4. Breast Cancer Cell Secretome-Mediated Chemoresistance and Recurrence

Breast cancer secretome-mediated chemoresistance is a complex process in which the secretome can create a supportive microenvironment for cancer cell growth and survival, leading to the development of drug resistance and, eventually, cancer recurrence. Post-therapy, various cells present in the TME, including CSCs, immune cells, fibroblasts, and endothelial cells, can be manipulated to promote tumor cell survival and relapse [122]. The secretome can also promote the survival of residual cancer cells that survive initial therapy. A common routine treatment for breast cancer patients with TNBC or high HER2 expression would be the administration of chemotherapy drugs as a neoadjuvant or adjuvant to surgery [123]. However, patients that relapsed with distant metastasis often develop chemoresistance with poor prognosis [122,124]. The development of chemoresistance can occur through two mechanisms, intrinsic or acquired resistance. Intrinsic resistance is caused by either inherent genetic mutations and/or by the heterogeneity and the protein interplay of the TME. In contrast, acquired resistance is by genetic modification, such as DNA damage repair or rewiring of intracellular signaling pathways during or after chemotherapy [51,125]. 

One of the key mechanisms of cancer cell secretome-mediated chemoresistance is the activation of the signaling pathway for survival and drug resistance in cancer cells. For example, cancer cells secrete IL-6 and TGF-β, which may encourage the expression of genes through autocrine or paracrine signaling. This includes genes that help to detoxify drugs (e.g., glutathione S-transferase P1 (GSTP1)), promote the activity of efflux pumps (e.g., multidrug resistance protein 1 (MDR1)), and regulate the effectiveness of chemotherapy to activate (e.g., STAT3 and SMAD) [76,77]. Alteration of the TME can also promote drug resistance. Secretion of ECM proteins such as collagen and fibronectin can create a dense matrix that limits drug penetration, creating a barrier that prevents chemotherapy drugs from reaching cancer cells. They can also induce the formation of abnormal blood vessels to further limit drug delivery [126]. Additionally, the breast cancer secretome components can induce the NF-κB pathway, which promotes cell survival by transcribing anti-apoptotic genes such as members of the B cell lymphoma 2 (Bcl-2) family and inhibitors of apoptosis (IAPs), as well as inhibits caspase cleavage [127]. 

Furthermore, the communication that occurs through EVs is becoming increasingly recognized as a significant factor in the development of chemoresistance in breast cancer. EVs have been found to carry anti-apoptotic proteins, Hsp 70, c-IAPs, and survivin [128]. Drug efflux pumps can be transported from chemoresistance cancer cells through EVs, either by directly packaging the functional protein or indirectly through encapsulation of mRNA [129]. When captured by chemosensitive cancer cells, this can result in effective drug efflux and subsequent chemoresistance. Findings indicate that EVs may also capture chemotherapeutic drugs, which could help to reduce drug toxicity. For example, EVs released by breast cancer cells are resistant to chemotherapy and can trap adriamycin [130].

## 4. Targeting the Breast Cancer Cell Secretome

Extensive studies of the breast cancer secretome have identified several potential targets for new cancer therapies, including antibodies or small molecules that can block the activity of specific proteins in the secretome [131,132]. In addition, analysis of the breast cancer secretome may also be useful for developing new biomarkers that can help for early detection and predict the likelihood of cancer recurrence or response to therapy. Therefore, it is imperative to develop novel therapeutic approaches to target secreted factors that are released into TME in order to prevent chemoresistance and relapse or enhance anti-tumor immunity. Several approaches have been used to target unique secretomes or constituents responsible for the secretion in breast cancer cells. For example, targeting the HER2 receptor using the FDA-approved monoclonal antibody trastuzumab to block its signaling pathway has been deemed efficacious for early and advanced breast cancer treatment [133]. Another FDA-approved alternative with the HER2 inhibitor lapatinib led to changes in the breast cancer cell secretome, promoting immune cell infiltration and activation [134,135]. The study found that lapatinib treatment led to an increase in the secretion of chemokines, such as CCL5 and CXCL10, that recruit immune cells, as well as an increase in the secretion of cytokines that activate immune cells, such as IFN-γ and TNF-α.

Another approach in targeting secretome is using drugs that block specific molecules’ secretion rate, such as IL-6, to reduce breast cancer proliferation. In breast cancer, the IL-6 signaling axis is a promising therapeutic target since it promotes growth and invasion, mediates the spread of metastatic capabilities, and is associated with poor prognosis [76,77]. Anti-IL-6 monoclonal antibodies such as sirukumab, olokizumab, MEDI5117, and clazakizumab have been used as inhibitors of the IL-6/JAK/STAT3 signaling pathway in various cancers. Still, the FDA has yet to approve these drugs for breast cancer treatment [77]. Reports also show that the secretome can be induced by targeted therapy with kinase inhibitors, resulting in significant alterations in the expressed secretome and enhanced drug resistance [136]. However, the study was performed in melanoma and lung cancer models, but the observation could be transferable to breast cancer. 

Targeting the epidermal growth factor receptor (EGFR) inhibitor in breast cancer with gefitinib was found to reduce the secretion of several proteins that promote tumor angiogenesis and invasion, including vascular endothelial growth factor (VEGF) and interleukin-8 (IL-8) [137]. The study also found that treatment with gefitinib increased the tumor suppressor protein thrombospondin-1 (TSP-1) secretion, which can inhibit tumor angiogenesis and promote apoptosis. Reports of other approved drugs targeting the secretome or TME components are listed in Table 1:

Breast cancer is a complex disease with a high recurrent rate. Drug combination therapy and precision medicine have emerged as promising strategies for improving treatment outcomes and reducing the risk of relapse [147,148,149]. The high relapse rate in breast cancer is caused by acquired resistance, which suggests the need for combination therapy [150]. Targeting one specialized microenvironment may lead to changes in other TME-related pathways because TME comprises numerous cells that frequently overlap and communicate. Therefore, a combination therapy targeting a specific microenvironment or niche may improve cancer treatment. Drug combination therapy involves using two or more drugs with different mechanisms of action to target multiple pathways involved in tumor survival and growth. This approach can improve treatment outcomes by enhancing the effectiveness of individual drugs, reducing the risk of drug resistance, and minimizing toxicity. Common practice would be the combination of chemotherapy and targeted therapy which has been shown to improve survival outcomes of patients compared to chemotherapy alone. For example, curcuminoid, a phenolic compound that has been utilized as a therapeutic agent in combination with chemotherapy, demonstrated enhanced efficacy in terms of reduced adverse effects and improved life quality in patients with solid tumors such as colorectal, gastric, and breast cancer in a phase II double-blind, randomized study [151,152,153,154].

The precision medicine approach would be tailoring treatment to the individual based on specific cancer characteristics, such as specific genetic mutations such as BRCA1/BRCA2 or the expression of specific biomarkers [155]. For example, the use of poly (ADP-ribose) polymerase (PARP) inhibitors has shown promising results in patients with BRCA mutations, which are associated with defects in DNA repair [156]. PARP inhibitors can block an alternative DNA repair pathway in these patients, leading to breast cancer cell death. In addition, emerging research suggests that combining precision medicine approaches with drug combination therapy may improve treatment outcomes and reduce the risk of relapse [157]. For example, combining a PARP inhibitor with a checkpoint inhibitor, pembrolizumab, enhances the immune system’s ability to target improved treatment outcomes in patients with BRCA-mutated breast cancer [156]. 

Understanding the landscape of breast cancer cell secretomes is essential for developing new cancer therapies. By targeting specific proteins involved in cancer cell signaling, researchers may be able to create more effective treatments that can slow or halt tumor growth. In addition, understanding the breast cancer cell secretome may also lead to the development of new diagnostic tools that can detect cancer earlier.

## 5. Conclusions

The field of breast cancer secretome research is currently undergoing a dynamic phase of investigation as scientists are exploring novel methodologies and tools to unravel the intricate mechanisms underlying breast cancer progression and metastasis. Recent advancements in high-sensitivity and high-throughput technologies, such as mass spectrometry-based proteomics/metabolomics, have revolutionized our ability to accurately identify and quantify the complex array of proteins, lipids, and metabolites that comprise the breast cancer secretome. Leveraging these cutting-edge analytical tools, researchers can now conduct a comprehensive and detailed analysis of the breast cancer secretome, enabling a deeper understanding of the pathophysiological processes involved in cancer development and progression. This could ultimately lead to the discovery of new therapeutic targets and the development of more effective treatments for breast cancer.

Microenvironmental alterations contribute to tumor progression and are attractive therapeutic targets, especially with metastatic breast cancer where outcomes are poor, necessitating novel treatment approaches. One of the therapeutic approaches is to enhance stromal-targeted therapy, but this requires careful clinical trial design, including neoadjuvant therapy. Nevertheless, targeting the microenvironment is a promising approach that could improve outcomes in breast cancer patients. It is evident that the role of secretome contributes to the hallmarks of cancer and intersects with major cancer-related pathways, as summarized in Figure 2. In addition, a recent study highlighted that genomic alterations induce changes in the TME components by changing the landscape of cancer cell fitness [158]. These TME components had distinct enrichment blueprints among breast cancer subtypes, with ER status exerting utmost influence, and some were associated with genomic profiles indicative of immune escape. In addition, the TME components also assert a prognostic impact on the clinical outcome of breast cancer patients. 

While targeting specific proteins in the breast cancer secretome can be effective, it is essential to note that breast cancer is a complex disease requiring a multifaceted treatment approach. Henceforth, should precision medicine approaches in the treatment of breast cancer occur, the unique characteristics of each patient’s genomic alteration and cancer cell secretome need to be considered. Therefore, continuous research into the breast cancer cell secretome and the role of TME is required to understand breast cancer progression further and develop new diagnostic and therapeutic strategies.

## Figures and Tables

**Figure 1 cancers-15-02653-f001:**
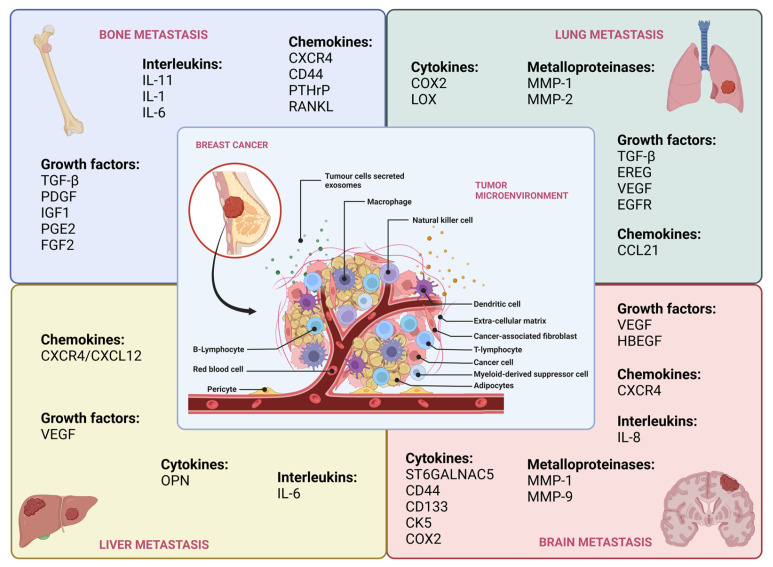
The summary of the signaling proteins that contribute to the site-specific metastasis in breast cancer.

**Figure 2 cancers-15-02653-f002:**
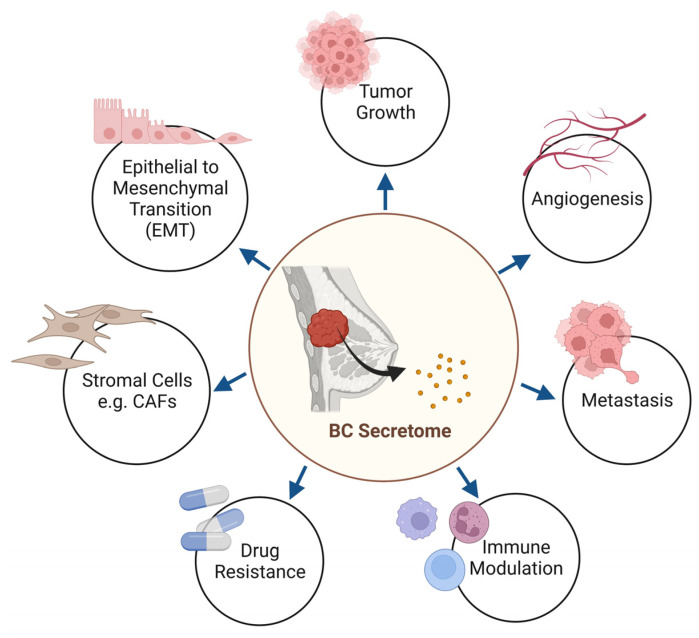
The breast cancer secretome contributes to the hallmarks of cancer.

**Table 1 cancers-15-02653-t001:** Reports of drugs targeting the components of TME.

Drug	Target	Molecular Target	Ref.
Bevacizumab	Angiogenic factor	VEGF	[138]
Lapatinib	Kinase inhibitor	EGFR	[139]
Ramucirumab	Angiogenic factor	VEGFR2	[140]
Pembrolizumab	Immune checkpoint inhibitor	PD-1	[141]
Anakinra	Interleukin	IL-1α, IL-1β	[142]
Canakinumab	Interleukin	IL-1β	[143]
Cetuximab	Kinase inhibitor	EGFR	[144]
Nivolumab	Immune checkpoint inhibitor	PD-1	[145]
Atezolizumab	Immune checkpoint inhibitor	PD-L1	[146]

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
