# Peer review of "Impact of the Cancer Cell Secretome in Driving Breast Cancer Progression"

_cancers, 2023, doi:10.3390/cancers15092653_

Round 1

Reviewer 1 Report

In this review, the authors Zahari et.al., have demonstrated the role of the secretome in breast cancer and how it interacts with tumor microenvironment and explores potential therapeutic opportunities. The review is well written and all the mechanisms are explained with the help of pictorial diagrams which is the strength of this review. I have just few suggestions to make:

Authors are suggested to explain the mechanisms of tumor cell secretome-mediated chemoresistance in a little more detail since it’s a very important and relevant concept.

Authors are suggested to include the mechanism behind cancer cell secretome dependent tumor recurrence.

Authors are suggested to include some examples of clinical trials and FDA approved drugs based on targeting components of cancer cell secretome.

Author Response

Responses to reviewer #1:

In this review, the authors Zahari et.al., have demonstrated the role of the secretome in breast cancer and how it interacts with tumor microenvironment and explores potential therapeutic opportunities. The review is well written and all the mechanisms are explained with the help of pictorial diagrams which is the strength of this review.

Response to Reviewer: We are grateful that the revisions were accepted by the respected reviewers. The suggestions have been very helpful in improving the manuscript.

Reviewer point #1: Authors are suggested to explain the mechanisms of tumor cell secretome-mediated chemoresistance in a little more detail since it’s a very important and relevant concept.

Response to Reviewer point #1: Thank you for the suggestion. We have added a new section 3.4 (line 431) explaining the possible mechanisms of tumor cell secretome-mediated chemoresistance.

Reviewer point #2: Authors are suggested to include the mechanism behind cancer cell secretome dependent tumor recurrence.

Response to Reviewer point #2: : Thank you for the suggestion. We agreed that the cancer recurrence is an important event to add in this review. We haved added a new section 3.4 combined with the reviewer point #1. Based on the literature, tumour  recurrence is multifactorial and is partly due to alteration of the extracellular matrix and immune cell infiltration caused by the secretome release, creating a supportive niche for cancer cells to grow and evade immune surveillance. This alteration in the microenvironment may contribute to breast cancer recurrence in which we have sub-divided them into specific subheadings 3.1-3.4.

Reviewer point #3: Authors are suggested to include some examples of clinical trials and FDA approved drugs based on targeting components of cancer cell secretome.

Response to Reviewer point #3: Thank you for the suggestion. We have actually dicussed some FDA-approved drugs in section 4, first paragraph line 538. This drug is mainly designed to target the breast cancer receptors which are also a component of the secretome. We have revised this section and  in addition to this, we have added a new Table 1 describing peer-reviewed articles reporting the drugs targeting the components of TME.

Reviewer 2 Report

The present review focuses on the impact of the secretome on breast cancer progression and how it interacts with the tumour environment and explores potential therapeutic opportunities targeting the secretome components. It involves 140 articles, with rich research content and large workload, which can provide readers with the latest progress in the field. Therefore, the article can be accepted after minor revision.

1. The content of the abstract and summary should be improved, which is less attractive to readers. It is suggested to appropriately expand the content and add hot keywords and innovation points.

2. In introduction: The review literature retrieval media, key words, etc. should be given. And clarify whether there are similar review articles before and the relationship with this article. To my knowledge, there have been some relevant review reports (JOURNAL OF EXPERIMENTAL & CLINICAL CANCER RESEARCH 2022, 41 (1), 203. SEMINARS IN CANCER BIOLOGY, 2020, 60 , pp.294-301. JOURNAL OF MAMMARY GLAND BIOLOGY AND NEOPLASIA, 2013, 18 , pp.267-275. JOURNAL OF PROTEOMICS, 2010, 73 , pp.1896-1906.)

3. Some references lack page information, such as 19, 54, 77, et al.

No problem.

Author Response

Responses to reviewer #2:

The present review focuses on the impact of the secretome on breast cancer progression and how it interacts with the tumour environment and explores potential therapeutic opportunities targeting the secretome components. It involves 140 articles, with rich research content and large workload, which can provide readers with the latest progress in the field. Therefore, the article can be accepted after minor revision.

Response to Reviewer: The suggestions provided by the reviewers have been instrumental in improving the manuscript, and we are deeply appreciative of their thoughtful and insightful feedback.

Reviewer point #1: 1. The content of the abstract and summary should be improved, which is less attractive to readers. It is suggested to appropriately expand the content and add hot keywords and innovation points.

Response to Reviewer point #1: Thank you for the suggestion. We have revised the layman summary and abstract and update the keywords and innovation points.

Reviewer point #2:  In introduction: The review literature retrieval media, key words, etc. should be given. And clarify whether there are similar review articles before and the relationship with this article. To my knowledge, there have been some relevant review reports (JOURNAL OF EXPERIMENTAL & CLINICAL CANCER RESEARCH 2022, 41 (1), 203. SEMINARS IN CANCER BIOLOGY, 2020, 60 , pp.294-301. JOURNAL OF MAMMARY GLAND BIOLOGY AND NEOPLASIA, 2013, 18 , pp.267-275. JOURNAL OF PROTEOMICS, 2010, 73 , pp.1896-1906.)

Response to Reviewer point #2: Thank you for the suggestion. The review only include indexed peer-reviewed article (mainly from PubMed repository) with special emphasis on articles <5 years old. Although there are similar articles, this review revisit the roles of TME in breast cancer pathogenesis. In addition there are increasing trends of these types of research and new view points that could help to understand the biology on breast cancer progression. As suggested, we have updated the aim of the review in the introduction.

Reviewer point #3. Some references lack page information, such as 19, 54, 77, et al.

Response to Reviewer point #2: Thank you for highlighting the error. We apologized since we are using a reference manager which may include error in retrieving the information. The missing information in the references and other references have been rectified.